# Investigating Causal Associations of Circulating Micronutrients Concentrations with the Risk of Lung Cancer: A Mendelian Randomization Study

**DOI:** 10.3390/nu14214569

**Published:** 2022-10-31

**Authors:** Haihao Yan, Xiao Jin, Linlin Yin, Changjun Zhu, Ganzhu Feng

**Affiliations:** 1Department of Respiratory Medicine, The Second Affiliated Hospital of Nanjing Medical University, Nanjing 210011, China; 2The Second Hospital of Nanjing, Nanjing University of Chinese Medicine, Nanjing 210046, China

**Keywords:** diet, nutrition, lung cancer, Mendelian randomization, causal risk factors

## Abstract

Previous observational studies have suggested that the effect of diet-derived circulating micronutrient concentrations on lung cancer (LC) risk is controversial. We conducted a two-sample Mendelian randomization (MR) analysis to investigate the causal relationship between circulating micronutrient concentrations and the overall risk of LC and three LC subtypes (namely lung adenocarcinoma (LA), squamous cell lung cancer (SqCLC), and small cell lung cancer (SCLC)). The instrumental variables (IVs) of 11 micronutrients (beta-carotene, calcium, copper, folate, lycopene, magnesium, phosphorus, retinol, selenium, zinc, and vitamin B6) were screened from the published genome-wide association studies (GWAS). Summary statistics related to LC and its subtypes came from the largest meta-analysis, including 29,266 cases and 56,450 controls. Inverse-variance weighted (IVW) method is used as the main MR analysis, and the sensitivity analysis is carried out to ensure the MR assumptions. This MR study found suggestive evidence that genetically predicted 6 circulating micronutrient concentrations was correlated with the risk of overall LC (odds ratio (OR): 1.394, 95% confidence interval (CI): 1.041–1.868, *p* = 0.026, phosphorus), LA (OR: 0.794, 95% CI: 0.634–0.995, *p* = 0.045, beta-carotene; OR: 0.687, 95%CI: 0.494–0.957, *p* = 0.026, calcium), SqCLC (OR: 0.354, 95% CI: 0.145–0.865, *p* = 0.023, retinol), and SCLC (OR: 1.267, 95% CI: 1.040–1.543, *p* = 0.019, copper; OR: 0.801, 95% CI: 0.679–0.944, *p* = 0.008, zinc). We found no evidence that other micronutrients are associated with the risk of overall LC or its subtypes. Our study suggested that the increase in circulating beta-carotene, calcium, retinol, and zinc concentration may reduce the risk of LC; the increase in circulating copper and phosphorus concentration may be related to the increased risk of LC. In the future, larger replication samples of LC genetic data and larger micronutrient-related GWAS will be needed to verify our findings.

## 1. Introduction

Lung cancer (LC) is a deadly tumor threatening human life and health. According to the GLOBOCAN assessment in 2020, the incidence of LC worldwide was 11.4%, ranking second among all new cancer cases. Its mortality accounted for 18.0% of all cancer-related deaths, making it the leading cause of cancer death [1]. Smoking is still the most significant risk factor for LC. Other nontobacco factors include air pollution, occupational exposure, chronic lung disease, and infection [2]. Presently, there are many disputes about the role of diet in the occurrence of LC. As epidemiological studies have found the benefits of vegetables and fruits, most health authorities recommend a balanced nutritional intake, including vegetables and fruits [3]. Early studies have shown that beta-carotene and vitamins A, C, and E may have protective effects on LC [4,5]. However, two subsequent large prospective trials yielded the opposite results. They found that the mortality rate in the group supplemented with beta-carotene and/or retinol was higher than that in the control group [6,7]. Other researchers have suggested that inadequate dietary intake of certain minerals, such as copper, iron, magnesium, and zinc, can lead to an increased risk of LC; their conclusion was also denied by a subsequent prospective cohort study, which found no association between total mineral intake and LC [8,9].

It is worth noting that most evidence about circulating vitamin and mineral concentrations and LC risk comes from observational studies [9,10,11]. These studies usually rely on the food frequency questionnaire (FFQ) to evaluate dietary intake, which is prone to measurement errors that lead to the misevaluation of the actual effect. As there may be differences in eating habits among individuals, it is difficult for observational studies to fully control all confounding factors. In addition, the timing, dose, and duration of micronutrient intake, as well as the uncertainty of the onset and long-term progression of LC, may explain the observed ineffective effects [9].

Mendelian randomization (MR) is a method to assess the causal association between risk factors and disease [12]. MR uses genetic variants as instrumental variables (IVs), which can effectively avoid the influence of confounding factors, which are difficult to control in observational studies [12]. Because the alleles affecting genetic variants are randomly assigned to offspring at conception and are not affected by environment and other unknown confounding factors, MR analysis is similar to natural randomized controlled trials, which can evaluate the causal relationship between exposure and outcome at the genetic level while ruling out the possibility of reverse causality [13].

In this study, we used a two-sample MR analysis to evaluate the causal correlation between genetically predicted circulating mineral and vitamin concentrations and the overall risk of LC and three LC subtypes, including lung adenocarcinoma (LA), squamous cell lung cancer (SqCLC), and small cell lung cancer (SCLC).

## 2. Materials and Methods

### 2.1. Study Design

Our study followed the newly produced STROBE-MR statement used to report MR research [14]. The potential causal effect of circulating micronutrient concentration on LC was studied by the two-sample MR method. MR is based on three critical assumptions: (1) IVs are closely related to circulating micronutrient concentration; (2) IVs should not be affected by known or unknown confounding factors; and (3) IVs only affect LC through circulating micronutrient concentration [15]. This study used publicly available data, so ethical approval and informed consent are unnecessary. Our study design is illustrated in Figure 1.

### 2.2. Selection of Genetic Instrumental Variables

We obtained the initial catalogue by searching Pubmed (https://www.ncbi.nlm.nih.gov/pubmed (accessed on 1 August 2022)) for published observational studies or meta-analyses of micronutrients related to LC. The preliminary catalogue includes 20 nutrients: beta carotene, calcium, copper, folate, iron, lycopene, magnesium, potassium, phosphorus, retinol, sodium, selenium, zinc, and vitamins B1, B2, B6, B12, C, D, and E [16,17,18,19,20,21,22,23,24,25,26,27,28,29]. We exclude iron and vitamins C, D, E, and B12 because the causal relationship between these circulating nutrient concentrations and LC risk has been thoroughly analyzed [30,31,32,33,34]. We further searched the genome-wide association studies (GWAS) catalogue (https://www.ebi.ac.uk/gwas (accessed on 4 August 2022)) and Pubmed for published GWAS on circulating micronutrient concentrations in European populations. Subsequently, potassium, sodium, and vitamin B1 and B2 were also excluded because no significant results of GWAS (*p* < 5 × 10^−8^) were found, or there was a lack of related studies. Finally, 11 nutrient-related GWASs were included in this study: beta-carotene, calcium, copper, folate, lycopene, magnesium, phosphorus, retinol, selenium, zinc, and vitamin B6 [35,36,37,38,39,40,41,42,43]. Single nucleotide polymorphisms (SNPs) were obtained from GWAS that independently affected the circulating micronutrient concentration in this study at a significant level (*p* < 5 × 10^−8^). These SNPs are not in linkage disequilibrium (r^2^ ≤ 0.1). We also excluded an SNP (rs6859667) representing selenium from the original catalogue because its minimum allele frequency (MAF) was less than 5% and could lead to the instability of the effect. To eliminate the influence of confounding factors, we tested the horizontal pleiotropy of each IV in the PhenoScanner database (http://www.phenoscanner.medschl.cam.ac.uk/ (accessed on 4 August 2022)). However, we did not find that SNPs included in this study were associated with confounding factors known to affect LC, such as smoking and alcohol consumption. F statistics are also used to evaluate weak IV bias in MR analysis. The F statistic was calculated using the following formula: F = R^2^ × (N − 2)/(1 − R^2^) [44], where N represents the sample size, and R^2^ refers to the variance of exposure explained by IVs [45]. When the calculated F statistic is more than 10, we think that the genetic variant used is a strong IV. To summarize, this study identified 39 SNPs related to 11 circulating micronutrient concentrations as IVs. Since part of the SNPs was not available in the outcome dataset in the MR analysis, we searched an online site (http://snipa.helmholtz-muenchen.de/snipa3/ (accessed on 4 August 2022)) to find the proxy-SNP (r^2^ > 0.8). Details about the selected SNPs and the agent SNP are shown in Appendix A.

### 2.3. Data on the Genetic Epidemiology of LC

Summary statistics on the association of selected exposure-related SNPs with LC and its subtypes come from a recent large meta-analysis [46]. The meta-analysis combined the genotypes of 14,803 cases and 12,262 controls from the OncoArray series with 14,436 cases and 44,188 controls from previous LC GWASs [47,48,49]. By comparing all study participants’ identity tags (IDs), the authors ensured no overlap between the current OncoArray dataset and the ATBC, CARET, and Eagle studies included in previous GWASs. Finally, we performed MR analysis of total LC (29266 cases and 56,450 controls) and subgroup analysis of LA (11273 cases and 55,483 controls), SqCLC (7426 cases and 55,627 controls), and SCLC (2664 cases and 21,444 controls).

### 2.4. Mendelian Randomization Analysis

After harmonizing the SNPs in the data source with the same allele, we performed the two-sample MR analysis. When only one SNP (such as lycopene) is available, the Wald ratio method is used to deduce the effect of a single IV on LC [12]. When the IVs consist of multiple SNPs, the fixed-effect inverse-variance weighted (IVW) method is used as the main method to estimate the effect of exposure on results [50]. This method is similar to a meta-analysis of the effects of a single SNP on the results, provides the most reliable causal estimation, and is relatively sensitive to pleiotropy [50]. Different MR methods were used for sensitivity analysis. Specifically, the weighted median approach (WMA) can provide a reliable causal estimate even if 50% of the IVs are invalid [51]. The MR-Egger method can detect potential pleiotropy and provide a more conservative estimate of causal effects after correcting the pleiotropy [52]. The existence of pleiotropy IVs may affect causal estimation. We first implemented Cochrane’s Q heterogeneity test to evaluate the degree of heterogeneity, where *p* < 0.05 indicates a high level of heterogeneity [53]. If heterogeneity is detected, the multiplicative random effect IVW method is used to avoid the deviation of heterogeneous IV association [54]. In order to assess the horizontal pleiotropy effect, the *p*-value of the MR-Egger regression intercept was used [55]. When the *p*-value < 0.05, it indicated that there was a high pleiotropy bias. Eventually, the MR pleiotropy residual sum and outlier (MR-PRESSO) method detects the existence of outlier IVs through the global test. It calculates the causal effect after removing the outlier IVs [56]. In addition, forest plots of MR analysis separately conducted by each SNP are provided.

Due to multiple tests, a bilateral *p* < 0.0045 (=0.05/11 outcomes) is set as the significant threshold. The *p*-value between 0.0045 and 0.05 is considered a suggestive correlation. We calculate the statistical power of the MR analysis using the specialized network tool mRND (https://shiny.cnsgenomics.com/mRND/ (accessed on 6 August 2022)). The power estimation of the circulating micronutrient concentration is based on a type I error of 5% [57]. The main parameters in the calculation included the sample size, case proportion, outcome odds ratio (OR), and R^2^. All data analysis in this study was implemented in R software (version 4.1.2). The R packages used include the TwoSampleMR and MR-PRESSO packages.

## 3. Results

### 3.1. Mendelian Randomization Estimates

Appendix A shows the specific characteristics of SNPs related to the circulating micronutrient concentration. The F statistics of all 39 SNPs are more than 10 (minimum = 16, maximum = 315), indicating no weak instrumental bias in our MR analysis. Specific information on the correlation of genetic variants with 11 circulating micronutrient concentrations and LC (including overall LC, LA, SqCLC, and SCLC) is shown in Appendix A.

The results of assessing the impact of circulating micronutrient concentrations on the risk of lung cancer using the Wald ratio method or IVW method are shown in Figure 2. Specifically, a suggestively positive association was observed for a standard deviation (SD) higher genetically predicted phosphorus level and overall LC risk (OR: 1.394, 95% CI: 1.041–1.868, *p* = 0.026). An SD higher concentration of beta-carotene and calcium was suggestively associated with a low risk of LA (OR: 0.794, 95% CI: 0.634–0.995, *p* = 0.045, beta-carotene; OR: 0.687, 95% CI: 0.494–0.957, *p* = 0.026, calcium). A suggestively negative association was observed for an SD higher genetically predicted retinol concentration and risk of SqCLC (OR: 0.354, 95% CI: 0.145–0.865, *p* = 0.023). In addition, a genetically predicted increase in the circulating copper concentration was suggestively associated with a higher SCLC risk (OR: 1.267, 95% CI: 1.040–1.543, *p* = 0.019), whereas a circulating zinc concentration was observed to be suggestively associated with a lower SCLC risk (OR: 0.801, 95% CI: 0.679–0.944, *p* = 0.008). We found no evidence that other micronutrients are associated with the risk of overall LC or its subtypes.

### 3.2. Evaluation of Mendelian Randomization Assumptions

For micronutrients with three or more SNPs, such as calcium, magnesium, phosphorus, selenium, and zinc, we used WMA and MR-Egger methods for the sensitivity analysis (Appendix A). In the sensitivity analysis, we found that the correlation modes remained directionally consistent in most statistical methods, and we also found a nominal correlation in the WMA of the circulating phosphorus concentration with overall LC (OR, 1.506; 95% CI, 1.007–2.253; *p* = 0.046) and the circulating zinc concentration with SCLC (OR, 0.790; 95% CI, 0.648–0.963; *p* = 0.020).

Cochrane’s Q test showed that the IVW method, as the primary MR analysis, was not affected by heterogeneity in all analyses (P_Cochrane’s Q_ > 0.05; Appendix A). Meanwhile, we found almost no evidence of horizontal pleiotropy in all analyses according to the MR-Egger intercept (P_intercept_ > 0.05; Appendix A). Finally, the MR-PRESSO global test did not recognize the existence of outlier SNPs (P_global test_ > 0.05; Appendix A). When we used the Wald ratio method to estimate and plot the forest plot of the effect of a single SNP for each micronutrient on the results, the results were consistent with the MR-PRESSO method (Appendix A). In addition, we have more than 80% statistical power to detect the significance of the relationship between beta-carotene (100%), calcium (84%), phosphorus (100%), retinol (100%), and LC. Meanwhile, we have nearly 80% of the statistical power to detect the association between copper (78%), zinc (71%), and LC (Appendix A).

## 4. Discussion

This study used summary statistics from genetic studies and large consortiums to investigate the causal relationship between 11 circulating micronutrient concentrations and LC. We found suggestive genetic evidence of a causal relationship between genetically predicted circulating beta-carotene, calcium, copper, phosphorus, retinol, and zinc concentrations and the overall risk of LC or its three subtypes.

The relationship between diet-derived micronutrients and the risk of LC has been controversial. Carotenoids and retinol have long been considered markers of vegetable and fruit intake. Researchers have closely watched them because of their anti-inflammatory and antioxidant properties that may have potential protective value for cancer [58]. Some studies have shown that increasing the dietary intake of beta-carotene and retinol helps to reduce the incidence of LC [4,5]. Other authors have found that dietary supplementation of beta-carotene and/or retinol increases the risk of death in LC patients [6,7]. Yu et al. [59] conducted a meta-analysis of 18 case-control studies. Their results showed that a higher dietary intake of beta-carotene was associated with a lower risk of LC. The association is particularly pronounced in Asian and American populations. Abar et al. [25] conducted a meta-analysis of 17 prospective studies. Their results showed that beta-carotene and retinol could significantly reduce the risk of LC occurrence and death. In the subsequent gender stratification analysis, researchers found that this effect is mainly reflected in men. However, a recent meta-analysis involving 167,141 participants yielded inconsistent results [18]. The meta-regression showed no relationship between the supplemental dose of beta-carotene and the size of the negative effect of lung cancer. The inconsistency of the above results may be because most of these studies rely on FFQs to evaluate dietary intake and are prone to measurement errors. Some studies also did not adjust for potentially important confounding factors, such as smoking and drinking, thus distorting the actual relationship between exposure and outcome. In addition, due to the design limitations of different studies, a meta-analysis is usually challenging to investigate the effects of micronutrients on different subtypes of LC, which may lead us to ignore the possible effects of nutrients on some subtypes of LC. Although our analysis did not prove the correlation between beta-carotene and retinol and the overall LC, the results of our subgroup analysis observed a suggestive association between beta-carotene and retinol and lung cancer subtypes. Specifically, circulating beta-carotene concentration may be correlated with reduced LA risk (OR: 0.794, 95% CI: 0.634–0.995, *p* = 0.045), whereas circulating retinol concentration may reduce the SqCLC risk (OR: 0.354, 95% CI: 0.145–0.865, *p* = 0.023). Our results suggest the potential benefits of dietary intake of vegetables and fruits in preventing LA and SqCLC.

The results are also inconsistent between calcium intake and the risk of LC. A meta-analysis by Yang et al. [19] combined the results of 32 studies and found no statistical correlation between dairy or calcium intake and LC risk. Nevertheless, the latest large meta-analysis [60] by Sun et al. found that calcium supplementation alone was not significantly associated with the risk of LC. However, calcium and vitamin D supplementation significantly reduced the incidence of LC, suggesting that calcium and vitamin D may synergistically affect tumor inhibition. Mechanism studies have found that signals from the 1, 25 (OH)_2_D_3_ receptor (VDR) and calcium-sensing receptor (CaSR) can inhibit tumor proliferation and metastasis and promote tumor differentiation and apoptosis [61,62]. Our results preliminarily suggest a suggestive association between calcium and LA risk reduction (OR: 0.687, 95% CI: 0.494–0.957, *p* = 0.026). Further research is required to clarify the role of vitamin D in calcium’s effect on LC.

There is little clinical study on phosphorus and LC. A Swedish study found that abnormal phosphorus levels in diet and serum were associated with the occurrence and development of various cancers [63]. Although the study found that estrogen plays a vital role in the connection between phosphorus metabolism and cancer, the results of the subgroup analysis of specific cancers showed that high phosphorus levels were positively correlated with LC risk in both men and women. We also found a multicenter retrospective study [64] that included 130 LC patients. The study found that the serum phosphorus level in patients with LC before treatment was higher than usual. In addition, serum phosphorus levels were negatively correlated with survival rates in LC patients. Pathophysiological evidence suggests that increased circulating phosphorus concentration can induce growth-promoting cell signal transduction, stimulate angiogenesis, and promote chromosome instability, leading to tumorigenesis. Animal experiments also found that high dietary phosphorus intake can lead to the growth of LC [65]. The results of our analysis also observed a suggestive causal correlation between phosphorus concentration and an increased risk of overall LC (OR: 1.394, 95% CI: 1.041–1.868, *p* = 0.026), which should be further evaluated in well-designed large-scale prospective studies.

Copper and zinc are important cofactors of many enzymes in the body, and they play an important role in maintaining the integrity of DNA [8]. Thus far, the role of copper and zinc in LC has been widely studied, and several meta-analyses have yielded similar results. Zhang et al. [20] performed a meta-analysis on 33 studies (including 3026 LC cases). Their results showed that the increase in circulating serum copper concentration was associated with the increased risk of LC. The results of a meta-analysis by Wang et al. [66] showed that the serum zinc level of LC patients was significantly inferior to the standard value. A recent meta-analysis used 39 observational studies to explore the relationship between the serum copper/zinc ratio and lung cancer [67]. The results showed that the serum copper/zinc ratio in LC patients was significantly higher than that in healthy controls and patients with benign lung diseases. Meanwhile, consistent results were observed in Asian and European populations. The results of our subgroup analysis further confirmed the suggestive causal effect of circulating copper (OR: 1.267, 95% CI: 1.040–1.543, *p* = 0.019) and zinc (OR: 0.801, 95% CI: 0.679–0.944, *p* = 0.008) concentrations in the development of SCLC. It is necessary to explore the potential carcinogenic mechanism of high levels of copper and low levels of zinc in the future.

This study’s main advantage is using data from genetic studies and large consortiums to assess the relationship between circulating micronutrients and LC, thus largely avoiding the common bias of observational studies. Secondly, the large sample size of MR analysis and the robust estimation effect of each SNP (F statistics > 10) ensure the statistical effectiveness of this study. We also searched the PhenoScanner database, largely avoiding the possibility that IVs might influence the results in other ways. In addition, we carried out a subgroup analysis to further explore the relationship between three subtypes of LC (LA, SqCLC, and SCLC) and micronutrients. Finally, sensitivity analysis was carried out to confirm the consistency of causal effects, and Cochrane’s Q, the MR-Egger intercept test, and the MR-PRESSO method were used to further eliminate the existence of pleiotropy.

Some possible limitations of this study need to be considered. First, changes in micronutrient concentrations may have diverse effects on lung cancer patients of different ages or gender. However, the lack of individual-level information in the summary statistics prevents us from stratifying LC by age and gender. Second, it should be noted that the current GWAS for LC does not include data on LC staging. Due to this limitation, it is difficult to determine whether micronutrient concentration is causally related to LC progression. Third, although we use various methods to avoid the impact of pleiotropy, we still cannot completely rule out the bias of unknown pleiotropy on the results. Fourth, the GWAS used in this study was based on the European population, so it is unclear whether our findings apply to the non-European population. Finally, due to the limitation of the sample size of the SCLC subgroup, the power of copper (78%) and zinc (71%) in our study did not reach 80%, which may affect our grasp of the significance of the results. Therefore, the effects of copper and zinc on SCLC should be carefully interpreted. In the future, we need a larger sample size of SCLC GWASs to prove our findings.

## 5. Conclusions

This study provided suggestive genetic evidence for a causal relationship between circulating micronutrient concentrations and LC risk. Specifically, the increased concentration of circulating beta-carotene, calcium, retinol, and zinc may reduce the LC risk; the increase in circulating copper and phosphorus may be associated with the increased risk of LC. In the future, larger replication samples of LC genetic data and larger micronutrient-related GWAS will be needed to verify our findings.

## Figures and Tables

**Figure 1 nutrients-14-04569-f001:**
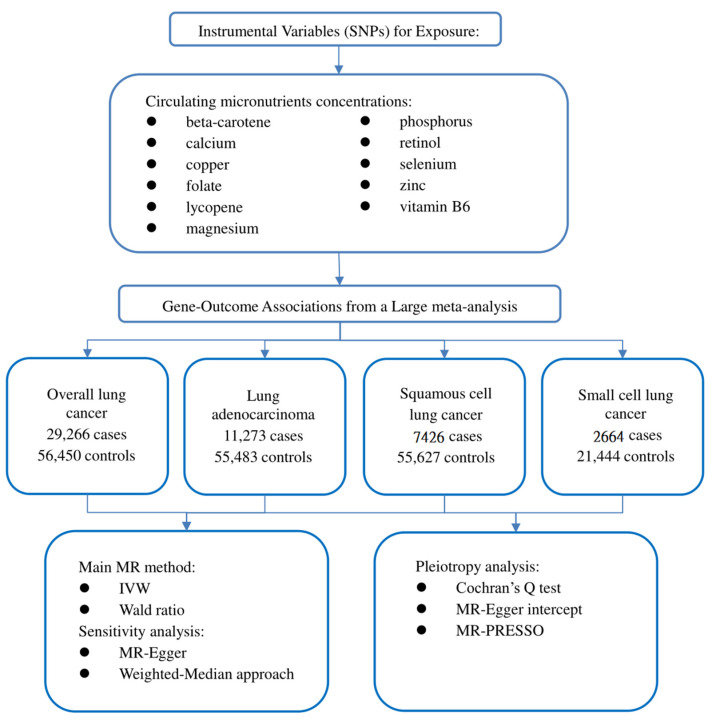
Flow chart of our study design.

**Figure 2 nutrients-14-04569-f002:**
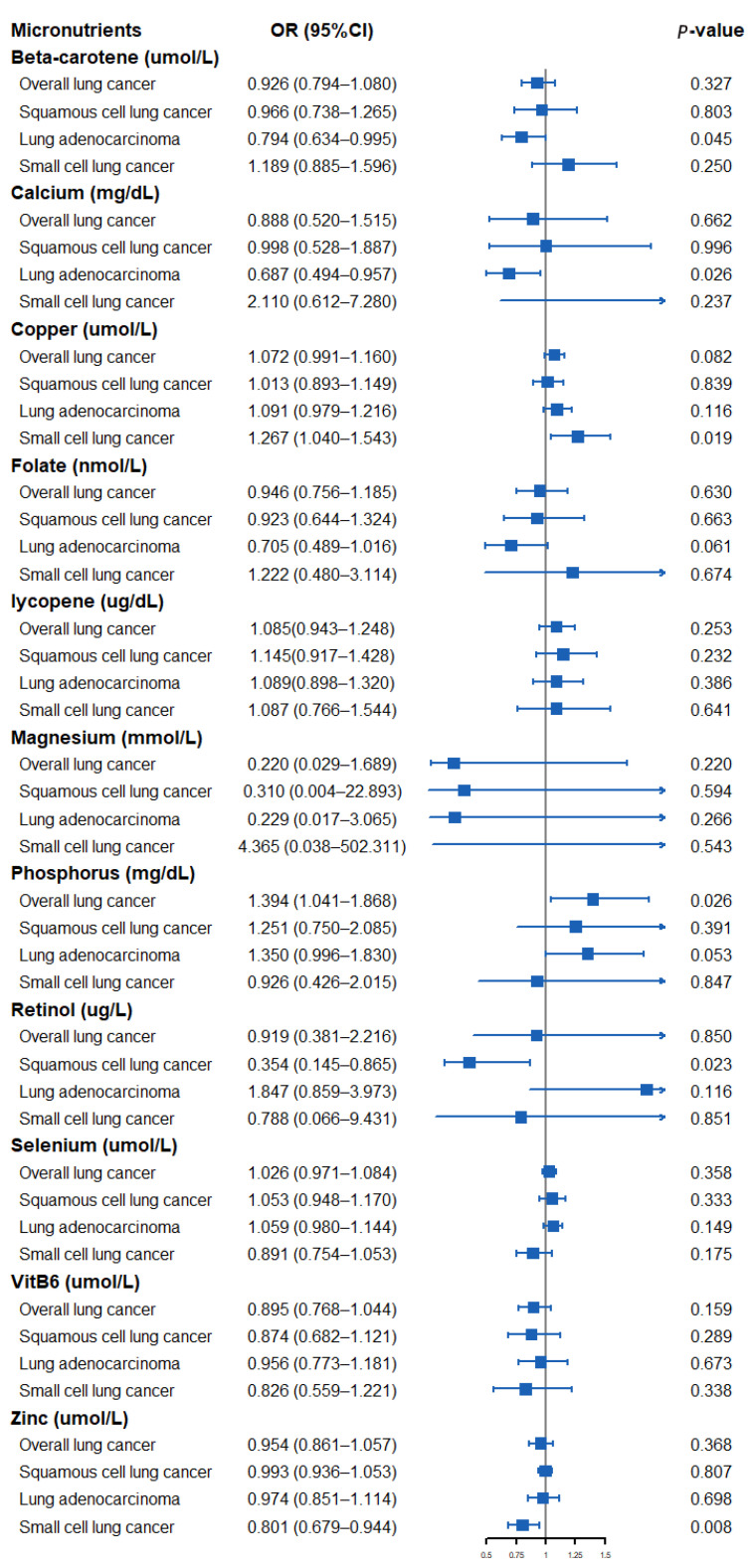
Forest plot showing results from Mendelian randomization study to assess potential causal associations between 11 micronutrients and lung cancer, including overall lung cancer, lung adenocarcinoma, squamous cell lung cancer, and small cell lung cancer.

## Data Availability

The GWAS data of lung cancer are accessible under application at https://www.ebi.ac.uk/gwas/ (accessed on 1 August 2022), and exposure data sources and handling of these data are described in Materials and Methods.

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
