# Peer review of "Investigating Causal Associations of Circulating Micronutrients Concentrations with the Risk of Lung Cancer: A Mendelian Randomization Study"

_nutrients, 2022, doi:10.3390/nu14214569_

Round 1
Reviewer 1 Report
Did the authors check if individuals with higher copper had Wilson’s disease or a family history of Wilson’s disease?
A limitation for this manuscript is that the gender of the individuals was not analyzed as a possible interaction. This would make the study more complete.
Why did the authors not evaluate any interaction of age with the differences in nutrients and cancers? Age is known to impact magnesium and zinc levels as well as other ions. The authors should address these areas of limitation.
Another issue is that the analysis did not take into account staging of the cancer along with the measurements and analyses. This should be addressed as differences in zinc or beta-carotene may be influenced by the general health of the subjects during the progression of cancer.
Why was small cell lung cancer omitted? Were there not a sufficient number of individuals for analyses?
The authors should check the references for the studies. Reference [16] title is sodium and potassium which was not analyzed.
Author Response
Point 1: Did the authors check if individuals with higher copper had Wilson’s disease or a family history of Wilson’s disease?
Response 1: The genetic variant associated with serum copper used in this study came from a genome-wide association study (GWAS). The study used two cohorts of adults from Australia and the UK. Participants were recruited from twins and their families in Australia and from pregnant women in the UK. Based on the information provided by the study, we found no family history of Wilson's disease or Wilson's disease in these participants. For more information, please refer to the reference "Evans DM, Zhu G, Dy V, et al. Genome-wide association study identifies loci affecting blood copper, selenium and zinc. Hum Mol Genet. 2013;22(19):3998-4006. doi:10.1093/hmg/ddt239".
Point 2: A limitation for this manuscript is that the gender of the individuals was not analyzed as a possible interaction. This would make the study more complete.
Response 2: Thank you for this suggestion. It would have been interesting to explore this aspect. However, the exposures and outcomes used in the Mendelian randomization (MR) study are based on the traits reported in the published summary statistics of GWAS, which do not include information at the individual level. Moreover, we did not find data to stratify lung cancer by gender in the GWAS catalog. Therefore, we cannot analyze the gender of the individuals as a possible interaction. Nevertheless, we elaborate on this limitation in the discussion. Please see the last paragraph of the discussion for details.
Point 3: Why did the authors not evaluate any interaction of age with the differences in nutrients and cancers? Age is known to impact magnesium and zinc levels as well as other ions. The authors should address these areas of limitation.
Response 3: Thank you for pointing this out. Although we agree that this is an important consideration, the purpose of this study was to explore the causal relationship between genetically predicted micronutrient concentration changes and the risk of lung cancer. The instrumental Variables (IVs) we chose are single nucleotide polymorphisms (SNPs) that are significantly associated with micronutrients, which itself represents the change in the level of circulating micronutrients. Therefore, we do not think it is necessary to consider the effect of age on the levels of magnesium and zinc. However, there may be differences in the effects of micronutrient concentrations on the risk of lung cancer at different ages. Unfortunately, the GWAS catalog also does not contain data to stratify lung cancer by age. According to your suggestion, we supplement this limitation in the discussion. Please see the last paragraph of the discussion for details.
Point 4: Another issue is that the analysis did not take into account staging of the cancer along with the measurements and analyses. This should be addressed as differences in zinc or beta-carotene may be influenced by the general health of the subjects during the progression of cancer.
Response 4: Thank you for pointing this out. We agree that as cancer progresses, the general health of the subjects may deteriorate and further affect zinc or beta-carotene levels. However, this belongs to the reverse effect of different stages of lung cancer on micronutrient concentration. Unfortunately, by searching the GWAS catalog, we found no summary statistics on the staging of lung cancer. Because of this limitation, we are unable to study the reverse causal relationship between different stages of lung cancer and micronutrient concentration. Meanwhile, it is difficult to analyze whether there is a causal relationship between micronutrient concentration and the progression of lung cancer. Therefore, it is difficult for us to modify according to your proposal, but we have emphasized this limitation in the discussion. Please see the last paragraph of the discussion for details.
Point 5: Why was small cell lung cancer omitted? Were there not a sufficient number of individuals for analyses?
Response 5: The effect of circulating micronutrient concentration on small cell lung cancer (SCLC) was not omitted in this study. Our results show that an increase in genetically predicted copper concentration is related to the increased risk of SCLC, and a genetically predicted increase in zinc concentration is related to the decreased risk of SCLC. We use the specialized network tool mRND (https://shiny.cnsgenomics.com/mRND/) to calculate the statistical power of MR analysis. Statistical power refers to how sure we are to believe in the significance of the results when p < 0.05. In general, the larger the sample size, the proportion of cases, and R2, the larger the power. When power is larger than 80%, we are more confident that the results are significant. However, due to the relatively small sample and case proportion of the SCLC subgroup, the power of copper (78%) and zinc (71%) in our study did not reach 80%, which may affect our grasp of the significance of the results. For more information about power, please refer to the manuscript “Brion MJ, Shakhbazov K, Visscher PM. Calculating statistical power in Mendelian randomization studies. Int J Epidemiol. 2013;42(5):1497-1501. doi:10.1093/ije/dyt179”.
Point 6: The authors should check the references for the studies. Reference [16] title is sodium and potassium which was not analyzed.
Response 6: We initially obtained the initial catalog and cited relevant reference, including sodium and potassium, by searching PUBMED for published observational studies or meta-analyses of micronutrients related to lung cancer. However, in a progressive analysis, we excluded potassium and sodium because no significant p-value (p < 5 × 10-8) was found. Therefore, we think that there is no error in the citation of reference [16].
Reviewer 2 Report
The authors performed two-sample Mendelian randomization (MR) analysis to investigate the causal relationship between circulating micronutrient concentrations and the overall risk of lung cancer (LC) and three LC subtypes, lung adenocarcinoma (LA), squamous cell lung cancer (SqCLC), and small cell lung cancer (SCLC). They found statistical evidence of the association between six circulating micronutrients and the risk of LC and its subtypes; the increase in circulating beta-carotene, calcium, retinol, and zinc concentrations may reduce the risk of LC, and the increase in circulating copper and phosphorus concentrations might increase the risk of LC.
Author Response
Point 1: The authors performed two-sample Mendelian randomization (MR) analysis to investigate the causal relationship between circulating micronutrient concentrations and the overall risk of lung cancer (LC) and three LC subtypes, lung adenocarcinoma (LA), squamous cell lung cancer (SqCLC), and small cell lung cancer (SCLC). They found statistical evidence of the association between six circulating micronutrients and the risk of LC and its subtypes; the increase in circulating beta-carotene, calcium, retinol, and zinc concentrations may reduce the risk of LC, and the increase in circulating copper and phosphorus concentrations might increase the risk of LC.
Response 1: We really appreciate your review of our manuscript and agree with our study.